# Pan-Genomics Reveals a New Variation Pattern of Secreted Proteins in *Pyricularia oryzae*

**DOI:** 10.3390/jof8121238

**Published:** 2022-11-23

**Authors:** Jiandong Bao, Zhe Wang, Meilian Chen, Shijie Chen, Xiaomin Chen, Jiahui Xie, Wei Tang, Huakun Zheng, Zonghua Wang

**Affiliations:** 1State Key Laboratory for Managing Biotic and Chemical Treats to the and Safety of Agro-Products, Institute of Plant Protection and Microbiology, Zhejiang Academy of Agricultural Sciences, Hangzhou 310021, China; 2Fujian Universities Key Laboratory for Plant Microbe Interaction, College of Life Sciences, Fujian Agriculture and Forestry University, Fuzhou 350002, China; 3Institute of Oceanography, Minjiang University, Fuzhou 350108, China; 4State Key Laboratory of Ecological Pest Control for Fujian and Taiwan Crops, College of Plant Protection, Fujian Agriculture and Forestry University, Fuzhou 350002, China

**Keywords:** *Pyricularia oryzae*, pan-genome, rice blast, population structure, secreted proteins

## Abstract

(1) Background: *Pyricularia oryzae,* the causal agent of rice blast disease, is one of the major rice pathogens. The complex population structure of *P. oryzae* facilitates the rapid virulence variations, which make the blast disease a serious challenge for global food security. There is a large body of existing genomics research on *P. oryzae*, however the population structure at the pan-genome level is not clear, and the mechanism of genetic divergence and virulence variations of different sub-populations is also unknown. (2) Methods: Based on the genome data published in the NCBI, we constructed a pan-genome database of *P. oryzae*, which consisted of 156 strains (117 isolated from rice and 39 isolated from other hosts). (3) Results: The pan-genome contained a total of 24,100 genes (12,005 novel genes absent in the reference genome 70-15), including 16,911 (~70%) core genes (population frequency ≥95%) and 1378 (~5%) strain-specific genes (population frequency ≤5%). Gene presence-absence variation (PAV) based clustering analysis of the population structure of *P. oryzae* revealed four subgroups (three from rice and one from other hosts). Interestingly, the cloned avirulence genes and conventional secreted proteins (SPs, with signal peptides) were enriched in the high-frequency regions and significantly associated with transposable elements (TEs), while the unconventional SPs (without signal peptides) were enriched in the low-frequency regions and not associated significantly with TEs. This pan-genome will expand the breadth and depth of the rice blast fungus reference genome, and also serve as a new blueprint for scientists to further study the pathogenic mechanism and virulence variation of the rice blast fungus.

## 1. Introduction

The rapid host adaptation of fungal pathogens represents a challenge in disease control [1]. The plant fungal pathogens often consist of diverged lineages colonized on different hosts, although some of these lineages are not strictly specialized and have overlaps in host ranges [2,3]. During the adaptation to different plant hosts, the fungal pathogens overcome the host immunity system either through the loss of genes coding for proteins which will activate the effector triggered immunity (ETI) [3,4], or the gain of genes coding for effectors required for the successful invasion of plant cells [5,6]. By alternating sexual and asexual reproduction, many plant fungal pathogens can overcome host resistance and achieve rapid host adaptation. In sexual reproduction, gene flow among different lineages facilitates the gain or loss of pathogenicity associated genes (PAGs) in the populations. In asexual reproduction, an absence of recombination ensures that those lineages with beneficial host adaptation loci are maintained and accumulated [7,8]. Deciphering the population structure of these pathogens will provide enhanced insights into mechanisms that drive rapid host adaptation, facilitate the timely prediction of disease pandemics, and deploy efficient management strategies.

A pan-genome contains all the gene information across all the strains of a species. In contrast to the comparative genomic analysis with the reference genome, it is advanced in obtaining the lineage-specific genes absent in the reference genome [9]. Genes in the pan-genome could be divided into two groups, the “core” genes and the “accessory” or “dispensable” genes, according to the gene frequency in the population [10]. Additionally, the pan-genome provides enhanced insights into the presence/absence variation (PAV)-based genome-wide association study (PAV-GWAS), establishing a possible association between the genotype and phenotype. To date, pan-genomes have been broadly used for population structure assessment, diversity analysis, and identification of important functional genes in humans [11] and plants [12,13,14,15,16,17,18], and also for the identification of novel pathogenic strains, or genes associated with infection in fungi [9,19,20,21] and bacteria [22]. 

*Pyricularia oryzae* (syn. *Magnaporthe oryzae*) is a complex species with different pathotypes infecting diverse grasses. It is most famous as the agent of the pandemic rice blast disease and the emerging wheat blast disease [23]. Lineages from different pathotypes are specialized, albeit not strictly, and are not pathogenic to other hosts. With the exception of the wheat-infecting strains with the Repeat-Induced Point mutation (RIP) mechanism [3,24], evidence supporting a naturally occurring sexual life cycle is still lacking for most of this species, although sexual reproduction could be observed under laboratory condition [2,25]. However, the rapid variations in avirulence genes (AVR) are common in the rice-infecting populations, which remains a considerable challenge in rice blast disease control strategies as most of the rice cultivars carry a single major resistance gene [26,27,28]. Increasing evidence reveals that transposable elements (TEs) are one of the major forces driving the variation of PAG genes in fungal pathogens, consequently promoting rapid pathogen adaptation to the host [28,29,30,31,32]. However, previous reference-genome-based studies mainly focused on the variation of conserved genes and shifted limited insights towards the population structure dynamics of *P. oryzae*. Here, we constructed a pan-genome consisting of 156 strains from different pathotypes and revealed a new variation pattern of secreted proteins in *P. oryzae*.

## 2. Results

### 2.1. Constructing the Pan-Genome of P. oryzae 

Based on the homologous gene clusters (OrthoFinder method), we constructed a rice blast fungus pan-genome that contains genome sequences from 156 strains (117 isolated from rice and 39 isolated from other hosts) from the NCBI repository (Figure 1A). The pan-genome contains 24,100 genes, nearly 1-fold (99.26%, 12,005 novel genes) more than the reference genome 70-15 (12,095 non-redundant genes). According to the frequency of pan-genes in the population, core genes (shared by ≥95% strains) account for 68.4% (half of which are hard-core genes present in all strains), strain-specific genes (only shared by <5% strains) account for 2.8%, and dispensable genes (frequency between 5% to 95%) account for 28.7% (Figure 1B). Among them, secreted proteins occupied a significant proportion (significant enrichment) of dispensable genes (Figure 1B).

As the gene increasing curve showed in Figure 1A, the first 117 rice-infecting strains have no overt difference, and the curve grows gently, while the curve from wheat-infecting strains and the other hosts increases abruptly, indicating that the genes in strains infecting different hosts except for rice are very different from each other. Gene presence (red)-absence (blue) variation (PAV) analysis results showed that ~70% of the pan-genes are core genes which present in almost all the strains (≥95%)(Figure 1C). This result is consistent with population frequency accounting (Figure 1B).

The gene PAV-based clustering analysis showed that all of tested *P. oryzae* strains consist of four clades (Figure 1D). 117 of rice-infecting strains clustered into three clades, containing 27, 33, and 57 strains, respectively. A total of 39 of the wheat or other host-infecting strains were clustered as Clade 4 (Figure 1D). This result is consistent with our previous results based on SNPs analysis of the *P. oryzae* population [33].

In order to facilitate the usage of these huge data generated in this study, we have built a website of the pan-genome Database of Rice Blast Fungus (http://47.107.41.214/, accessed on 22 November 2022), which is an open access resource. The website provides functions including ID or sequence similarity search, data browser such as homologous gene clusters, gene presence and absence polymorphisms, information on the strains used in this study, and download of sequences of the pan-genome.

### 2.2. Functional Enrichment Analysis of Pan-Genes

Functional enrichment analysis found that different types of pan-genes in *P. oryzae* performed different function preferences. The strain-specific genes are mainly involved in the individual adaptation and response to the changing environment, like Photosynthesis, Cytochrome P450, and Cellulose synthase (Figure 2A,B), while the core genes preferably regulate basic internal cellular processes, such as protein/ATP/FAD binding and sugar (and other) transporter (Figure 2C,D).

### 2.3. Functional Enrichment Analysis of Newly Identified Genes in Pan-Genome

Compared with the reference genome 70-15 (12,095 unique genes), the pan-genome of 117 strains from the rice host contains a total of 22,882 genes and revealed 10,787 new genes (~90%) absent in the 70-15 genome (Figure 3A). Further analyses showed a significant enrichment of genes coding secreted proteins in these novel genes which were significantly associated with transposable elements (TE) (Figure 3B). These novel genes from rice-infecting strains are mainly involved in the regulation of alcoholism, Glycolysis/Gluconeogenesis and IL-17 signaling pathway in KEGG annotation, and genes encoding tropomyosin, such as RNase H and Photosynthetic reaction center proteins in Pfam annotation (Figure 3C).

Compared with the pan-genes in rice-infecting strains, strains from other hosts had a total of 23,139 genes, including 1218 new genes (Figure 3A). Likewise, these novel genes from non-rice host strains were also enriched for secreted proteins and significantly associated with TE (Figure 3B). Functionally, these novel genes are enriched in KEGG pathways mainly involved the regulation of oxidation-reduction processes, enoyl-[acyl-carrier-protein] reductase (NADH) activity, and ADP binding, as well as being enriched in Pfam annotations, such as the cytochrome P450 family, Carboxylesterase family, and some conserved hypothetical proteins (Figure 3D).

### 2.4. Functional Analysis of Population Differentiation Related Genes in the Pan-Genome

Through the analysis of the subgroup-specific genes (SSGs) in the rice host strains, we found that the subgroups Clade 1, Clade 2, and Clade 3 have 195, 165, and 296 SSGs, respectively, which were present only in selected subgroups, but absent in other subgroups (Figure 4A). Gene functional association analysis showed that there is not a significantly association between SSGs and pathogen–host interaction-related genes (PHI), TE and secreted proteins, except that SSGs in Clade 1 have a significant association with TE (6.03 × 10^−8^) (Figure 4B). The KEGG pathway enrichment analysis showed that these genes were mainly enriched in pathways like lysine biosynthesis, starch and sucrose metabolism, and phenylpropane biosynthesis (Table 1).

Meanwhile, for the core genes under selection (CGUS) 117 rice-infecting strains were also selected as the population differentiation candidate genes. According to the values of F_st_ and Tajima’s D (F_st_ > 0.25, Tajima’s D < 0), 576 (Clade 1 vs. Clade 2), 620 (Clade 1 vs. Clade 3), and 697 (Clade 2 vs. Clade 3) CGUS were obtained (Figure 4B). All of them were significantly associated with the PHI, but not significantly associated with TE and secreted proteins (Figure 4B). Pfam enrichment analysis showed that these genes mainly enriched in the phospholipase D active site motif, E1-E2 ATPase, and ubiquitin carboxyl-terminal hydrolase (Table 2).

### 2.5. Pan-Genomics Reveals a New Variation Pattern of Secreted Proteins in P. oryzae

We calculated the population frequencies of the nine cloned avirulence (AVR) genes in *P. oryzae* and found that most of the AVR genes belong to the core genes (Figure 5A), instead of what we usually consider variable genes. *AVR1-CO39*, *AVR-Pi54*, *AvrPi9* and *AvrPiz-t* are 100% present in all of the collected strains, except for *AvrPii* and *AvrPia*, which are present in only a few strains. In order to verify the distribution of effector proteins, we counted the gene frequencies of all predicted secreted proteins in the rice-infecting strains and found a similar trend in the secreted proteins enriched in high-frequency core genes or low-frequency strain-specific genes (Figure 5B). Further analysis found that conventional secreted proteins (with signal peptides) were enriched in high-frequency near-core genes (*p* = 4.05 × 10^−37^) and were significantly associated with TE (*p* = 4.90 × 10^−18^). Meanwhile, unconventional secreted proteins (without signal peptides) were enriched in the low-frequency strain-specific genes (*p* = 4.34 × 10^−44^) but were not significantly associated with TE (Figure 5C).

## 3. Discussion

In this study, we constructed a pan-genome of *P. oryzae* using 156 genome assemblies, including 117 rice-infecting strains and 39 strains from other hosts. The pan-genome consists of 24,100 genes, which is about two times of that of the reference genome 70-15. Consistent with previous studies in fungi [9,19,20,34], core genes constitute the major component of the *P. oryzae* pan-genome (~70%). The newly identified genes enriched in secreted proteins as well as proteins involved in diverse cellular processes. In contrast to the 70-15 reference genome, some of the field isolates contain either supernumerary chromosomes (accessory chromosome or lineage-specific chromosomes), including the wheat blast isolate B71, rice blast isolates Y34 and FR13, and the grass isolate TF05-1 [35,36,37], or lineage-specific regions [3,38,39]. These isolate-specific genetic components often encode unique genes involved in the interaction with their respective host plants, and therefore may largely contribute to the dynamic pan-genome, as well as the host adaption of *P. oryzae*. 

In addition to the other environmental factors [2], host selective stress is the major force driving the specialization of pathogens [25,40]. In this study, the PAV of genes subdivided the isolates into four Clades, which is consistent with the previous result from our lab using whole genome SNP [33]. The gain and loss of effector gene function are essential for the host adaptation of different pathotypes within a species [4,41,42]. For example, the *P. oryzae* isolate that caused the outbreak of wheat blast disease in Brazil was supposed to be derived from the host jump of a Lolium isolate through the loss of function of the PWT3 gene [3]. We have previously found that all the cloned AVR genes in *P. oryzae* were absent in the *P. penniseti* isolate P1609 [43]. However, our results in current study suggested a ubiquitous distribution of the cloned AVR genes in all the isolates investigated, except for *AvrPii* and *AvrPia* which are exclusively present in only a few isolates. The discrepancy could be caused by the isolates used in this study.

TEs are often enriched in the fast-evolving compartments of fungal pathogen genomes and are associated with variations in chromosome structure and gene function, such as deletion, insertion, or expression [31,44,45]. Therefore, the localization of PAGs in proximity to TEs therefore constitutes a rapid host adaptive evolution mechanism for the fungal pathogens [30,46,47]. Here, we showed that both the core and accessory secreted proteins in the pan-genome are associated with TE, which is in agreement with our previous study [29]. Importantly, we further compared the distribution of secreted proteins with and without N-terminal signaling peptides (conventional and unconventional SPs). We found that the former was enriched in the high-frequency region and localized proximately to TEs, and the latter was enriched in the low-frequency region and not associated with TEs. We inferred from these results that unconventional SPs without N-terminal signaling peptide might employ distinct variation mechanism compared with conventional SPs with a signal peptide. 

Although only those genomes with 95% completeness were used in this study, the quality of this pan-genome was insufficient for scrutinizing additional features, such as the TE insertion around effector genes. Furthermore, more complete genomes derived from long reads sequencing data are necessary to produce deeper insight into the dynamic pan-genome structure and the rapid adaptive evolution of this notorious fungal pathogen. 

## 4. Materials and Methods

### 4.1. Genomics Data Collection

A total of 199 genome assembies of *P. oryzae* were collected (Appendix A) from the NCBI assembly database (https://www.ncbi.nlm.nih.gov/genome/browse/#!/eukaryotes/62/, accessed on 30 December 2019). Of those, over 50% were submitted by our group (Prof. Zonghua Wang’s lab in Fujian Agriculture and Forestry University). The quality of these genome assemblies was assessed by BUSCO v4.1.2 [48] with core orthologs at sordariomycetes level (*n* = 3817), and a total of 156 genome assemblies from different strains with BUSCO completeness greater than 95% were selected for pan-genome analysis.

### 4.2. Gene Prediction 

The 156 genome assemblies were repeat masked by RepeatMasker v4.1.0 (http://www.repeatmasker.org/, accessed on 1 June 2021), and then used for genes prediction by the software fgenesh (species = *Magnaporthe oryzae*) in MolQuest v2.4.5 (http://molquest.com, accessed on 10 January 2021).

### 4.3. Pan-Genome Construction

All protein sequences of the 156 strains were clustered together using OrthoFinder v2.5.1 [49] to obtain the homologous gene clusters. Those strain-specific unclustered genes and representative genes in each gene clusters were selected to construct a pan-genome. 

### 4.4. Presence/Absence Variations (PAVs) of Pan-Genes

PAVs of pan-genes in the population were determined by sequence similarity search conducted by NCBI-BLAST tools against genome sequence of each strain. According to the gene frequency in the population, genes were divided into core genes (≥95%), strain-specific genes (≤5%) and dispensable genes (5%–95%).

### 4.5. Clustering Analysis of Population Structure 

The R package clusterProfiler [50] was used for population structure analysis, the strains of rice blast fungus were divided into different subgroups based on PAVs.

### 4.6. Gene Functional Annotation and Enrichment Analysis 

KEGG (Kyoto Encyclopedia of Genes and Genomes) was annotated by with KofamKOALA (https://www.genome.jp/tools/kofamkoala/, accessed on 30 June 2021), KOG was annotated by eggNOGMapperV2 (http://eggnog-mapper.embl.de/, accessed on 30 June 2021), Pfam and GO were annotated by InterProScan v5.55-88.0 (https://www.ebi.ac.uk/interpro/, accessed on 30 June 2021), pathogenicity-related genes were annotated by PHI-base (http://www.phi-base.org/, accessed on 30 June 2021). ClusterProfiler [50] was used for gene functional enrichment analysis.

### 4.7. Population Differentiation Related Genes Analysis

For the core genes present in relevant subgroup, we used MAFFT (https://mafft.cbrc.jp, accessed on 1 August 2021) to perform multiple sequence alignment of the genome sequences corresponding to the homologous gene clusters, and then used the R package PopGenome [51] to calculate Tajima’s D and F_st_. Genes with Fst > 0.25 and Tajima’s D < 0 were selected as candidate population differentiation related genes.

For the unique genes of each subgroup, we selected the genes conserved in a certain subgroup but absent in other subgroups as candidates of population differentiation related genes.

## Figures and Tables

**Figure 1 jof-08-01238-f001:**
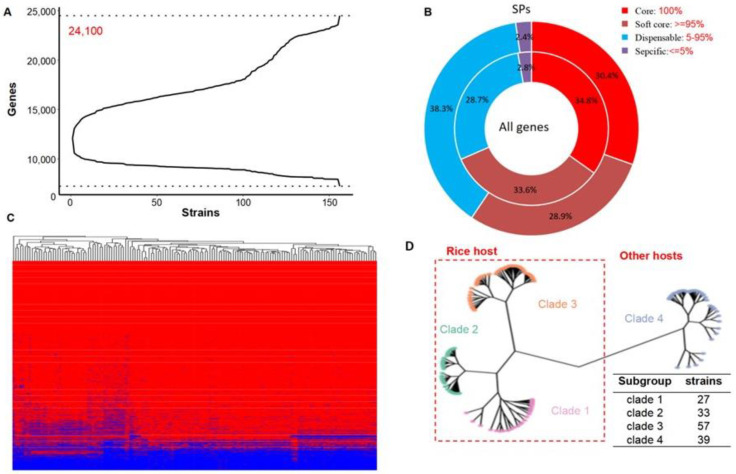
Profile of the pan-genome of *P. oryzae*. (**A**) The curve of pan-genome. (**B**) Type and percentage of pan-genes. SPs: secreted proteins. (**C**) Gene presence (red)-absence (blue) variation (PAV) analysis. (**D**) Gene PAV-based clustering analysis. It can be divided into four subgroups, including three subgroups from the rice host and one other subgroup from other hosts like wheat.

**Figure 2 jof-08-01238-f002:**
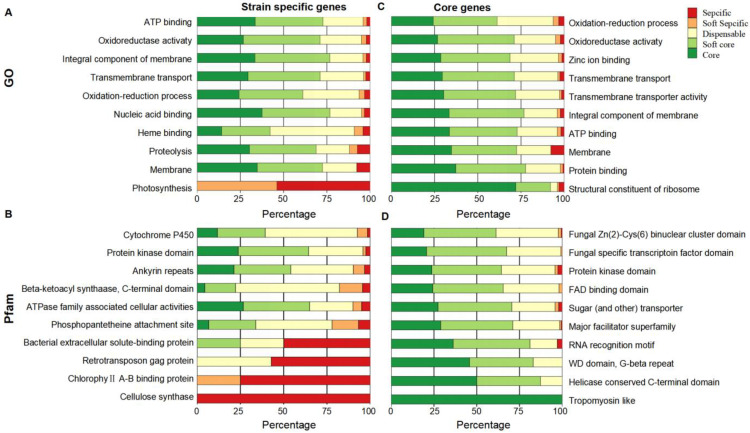
Functional enrichment analysis of different types of genes. (**A**) GO and enrichment analysis of strain-specific genes. (**B**) Pfam enrichment analysis of strain-specific genes. (**C**) GO enrichment analysis of core genes. (**D**) Pfam enrichment analysis of core genes.

**Figure 3 jof-08-01238-f003:**
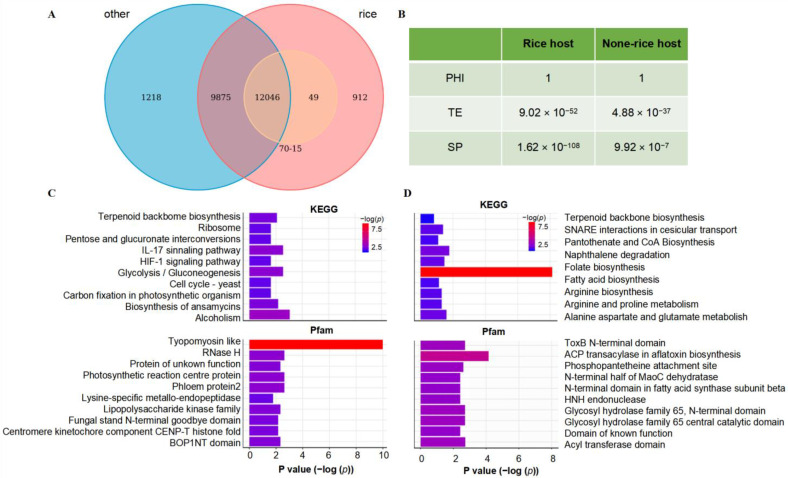
Functional enrichment analysis of new genes in pan-genome. (**A**) Comparative analysis of pan-genes from different hosts and the reference genome 70-15. (**B**) Association analysis of the new pan-genes from different hosts with Pathogen Host Interactions (PHI), TE and secreted proteins. (**C**) Functional enrichment analysis of new genes in rice-infecting strains. (**D**) Functional enrichment analysis of new genes in non-rice-infecting strains.

**Figure 4 jof-08-01238-f004:**
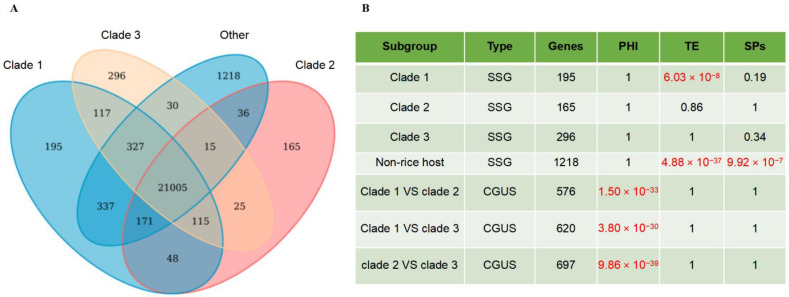
Population differentiation related genes in the pan-genome. (**A**) Comparison of the differences between different subgroups. (**B**) Functional enrichment analysis of population differentiation related genes. SSG: Subgroup Specific Genes; CGUS: Core Genes Under Selection.

**Figure 5 jof-08-01238-f005:**
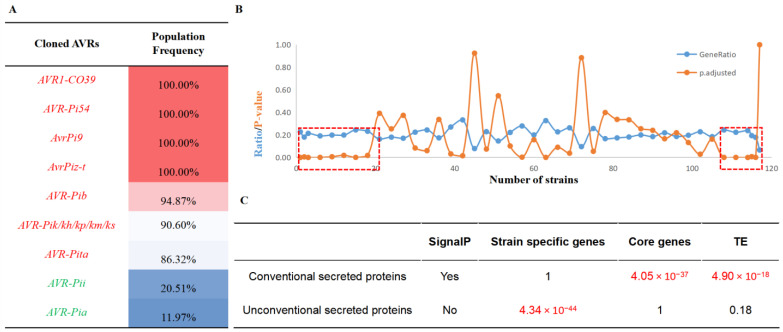
The secreted proteins variation patterns in the pan-genome. (**A**) The population frequency of cloned avirulence genes. (**B**) The population frequency of secreted proteins. It showed that secreted proteins were significantly enriched in the regions of core genes and specific genes. (**C**) Conventional secreted proteins were enriched in core genes and have significant association with TE. Meanwhile, the unconventional secreted proteins were enriched in specific genes and have no significant association with TE.

**Table 1 jof-08-01238-t001:** The KEGG pathway enrichment analysis of SSGs in rice-infecting strains (Top3).

Type	Enriched KEGG Pathway	*p*-Value
Clade 1	Lysine biosynthesis	1.07 × 10^−2^
Clade 1	Drug metabolism—other enzymes	1.07 × 10^−2^
Clade 1	Sulfur metabolism	2.12 × 10^−2^
Clade 2	Starch and sucrose metabolism	7.33 × 10^−3^
Clade 2	Glyoxylate and dicarboxylate metabolism	7.33 × 10^−3^
Clade 2	Nitrogen metabolism	7.33 × 10^−3^
Clade 3	Carbon fixation in photosynthetic organisms	1.94 × 10^−3^
Clade 3	Hepatocellular carcinoma	6.72 × 10^−3^
Clade 3	Phenylpropanoid biosynthesis	1.42 × 10^−2^

**Table 2 jof-08-01238-t002:** The Pfam enrichment analysis of the core genes in rice-infecting strains.

Type	Pfam ID	Pfam Description	*p*-Value
clade2 vs. clade3	PF00614	Phospholipase D Active site motif	3.68 × 10^−4^
clade2 vs. clade3	PF00122	E1-E2 ATPase	4.53 × 10^−4^
clade1 vs. clade2	PF00443	Ubiquitin carboxyl-terminal hydrolase	5.54 × 10^−4^
clade1 vs. clade2	PF00082	Subtilase family	6.69 × 10^−4^
clade1 vs. clade3	PF04082	Fungal specific transcription factor domain	1.09 × 10^−3^
clade1 vs. clade3	PF00082	Subtilase family	1.18 × 10^−3^
clade2 vs. clade3	PF00454	Phosphatidylinositol 3- and 4-kinase	1.46 × 10^−3^
clade2 vs. clade3	PF00702	haloacid dehalogenase-like hydrolase	2.16 × 10^−3^
clade1 vs. clade3	PF06422	CDR ABC transporter	2.33 × 10^−3^
clade1 vs. clade3	PF14510	ABC-transporter N-terminal	2.33 × 10^−3^

## Data Availability

All the data reported in this study could be found at the website of the Pan-genome Database of Rice Blast Fungus (http://47.107.41.214/).

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
