# Peer review of "Pan-Genomics Reveals a New Variation Pattern of Secreted Proteins in Pyricularia oryzae"

_jof, 2022, doi:10.3390/jof8121238_

Round 1
Reviewer 1 Report
Review Bao et al. 2022
The paper describes a data base that reports presence-absence polymorphisms within a large series of 156 genomic sequences from Magnaporthe oryzae.
The work is interesting, scientifically sound a may be a useful resource for scientists in the field.
However the writing is not good enough at this stage to allow publication.
Many sentences are not English and others remain unclear even to the experts in the field.
Specific comments:
L24: what does 1 fold mean?
L27: grammar
L29: not clear
L44: not english
L46 not english
L58 : not english
L77: unclear
Many other places.
Author Response
Response to Reviewer 1 Comments
The paper describes a data base that reports presence-absence polymorphisms within a large series of 156 genomic sequences from Magnaporthe oryzae. The work is interesting, scientifically sound a may be a useful resource for scientists in the field.
Point 1: However the writing is not good enough at this stage to allow publication. Many sentences are not English and others remain unclear even to the experts in the field.
Response 1: Thanks for pointing out it. we have carefully gone through the manuscript, and fixed writing issues with the help from a native English-speaking collaborator (Dr. Norvienyeku Justice, associate professor at the school of Plant Protection, Hainan University).
Specific comments:
Point 2: L24: what does 1 fold mean?
Response 2: Pan-genome of Pyricularia oryzae (24,100 genes) revealed 12,005 novel genes absent in reference genome 70-15 (12,095 genes), 12,005 novel genes is around 1 fold of that in reference genome 70-15. To avoid misunderstanding, we have revise in L24 as ”The Pan-genome contained a total of 24,100 genes (12,005 novel genes absent in the reference genome 70-15)”.
Point 3: L27: grammar
Response 3: Thank you for pointing out it. We have addressed as follow:
”Gene presence-absence variation (PAV) based clustering analysis of the pupolation structure of P. oryzae revealed four subgroups (3 from rice and 1 from other hosts).”
Point 4: L29: not clear
Response 4: Thank you for pointing this out. We have revised as follow:
”Interestingly, the cloned avirulence genes and conventional secreted proteins (SPs, with signal peptides) were enriched in the high-frequency regions and significantly associated with transposable elements (TEs); While the unconventional SPs (without signal peptides) were enriched in the low-frequency regions and not associated significantly with TEs.”
Point 5: L44 and L46: not english
Response 5: Thank you for pointing this out. We have revised as follow:
”By alternating sexual and sexual reproduction, many plant fungal pathogens get the best of host adaptation. In the sexual reproduction, gene flow among different lineages facilitates the gain or loss of pathogenicity associated genes (PAGs) in the populations. In the asexual reproduction, an absence of recombination ensures that those lineages with beneficial host adaptation loci are maintained and accumulated [7, 8]. ”
Point 6: L58 : not english
Response 6: Thank you for pointing this out. We have revised as follow:
”Also, pan-genome provides enhanced insights into the presence/absence variation (PAV)-based genome-wide association study (PAV-GWAS), establishing a possible association between the genotype and phenotype. ”
Point 7: L77: unclear
Response 7: Thank you for pointing this out. We have revised as follow:
”However, previous reference genome based studies mainly focused on the variation of conserved genes, and shifted limited insights into the population structure dynamics of P. oryzae”
Point 8: Many other places.Language concern:
Response 8: Thank you for your comments. With help from Dr. Norvienyeku Justice (a native English-speaking collaborator, associate professor at the school of Plant Protection, Hainan University), we have carefully gone through the manuscript, and fixed writing issues. Details see our revised manuscript.
Reviewer 2 Report
I checked your manuscript and described comments below.
Rice blast is a serious disease affecting rice yields.
The superior points of this paper are as follows.
1. In this research, I think it is very good that the author independently created the Pyricularia oryzae Pan-genome Database.
2. This paper clarified that ``P. oryzae could be divided into 4 subgroups (3 from rice and 1 from other hosts), and the subgroup-specific genes were less than 5%.''
3. the cloned avirulence genes and conventional secreted proteins (SPs) with signal peptides were enriched in the high-frequency regions and were significantly associated with transposable elements (TEs)
I don't think this paper has any major mistakes or grammatical problems.
Author Response
Response to Reviewer 2 Comments
I checked your manuscript and described comments below.
Rice blast is a serious disease affecting rice yields.
The superior points of this paper are as follows.
- In this research, I think it is very good that the author independently created the Pyricularia oryzaePan-genome Database.
- This paper clarified that “P. oryzae could be divided into 4 subgroups (3 from rice and 1 from other hosts), and the subgroup-specific genes were less than 5%.”
- the cloned avirulence genes and conventional secreted proteins (SPs) with signal peptides were enriched in the high-frequency regions and were significantly associated with transposable elements (TEs)
I don't think this paper has any major mistakes or grammatical problems.
Response : Thank you very much for your high appreciation of our work and positive comments. We appreciate the time and effort that you spend on providing feedback on our manuscript.